# A Systematic Review of Randomized Clinical Trials Evaluating the Efficacy of Minimally Invasive Surgery for Soft Tissue Management: Aesthetics, Postoperative Morbidity, and Clinical Results

**DOI:** 10.3390/medicina59050924

**Published:** 2023-05-11

**Authors:** Carlos M. Ardila, Daniel González-Arroyave, Annie Marcela Vivares-Builes

**Affiliations:** 1Basic Studies Department, School of Dentistry, Universidad de Antioquia UdeA, Medellín 050010, Colombia; 2Medicine Department, Hospital San Vicente Fundación, Rionegro 054047, Colombia; daniel.gonzalez@sanvicentefundacion.com; 3School of Dentistry, Institución Universitaria Visión de Las Américas, Medellín 050031, Colombia; anny.vivares@uam.edu.co

**Keywords:** minimally invasive surgery technique, clinical trials, efficacy, aesthetic, postoperative morbidity, systematic review

## Abstract

*Background and Objectives*: The necessity for less invasive and patient-friendly surgical therapies guided the development of the “minimally invasive surgical technique” (MIST). The aim of this systematic review was to evaluate the efficacy of MIST for soft tissue management considering aesthetic results, postoperative morbidity, and clinical outcomes. *Materials and Methods*: Several databases were used to conduct a thorough analysis of the scientific evidence. To investigate randomized clinical trials (RCTs), MeSH terms and keywords were provided. *Results*: Eleven RCTs were chosen. These experiments included 273 patients. The trials that explored MIST for papilla preservation presented greater efficacy in increasing papillary height (*p* < 0.05). MIST showed stable clinical outcomes for the management of excessive gingival display and with a flapless technique for single implant placement. Considering the treatment of gingival recessions, some RCTs presented greater root coverage with MIST (*p* < 0.05), while other trials did not show differences between groups. Regarding aesthetic perception, five RCTs indicated high patient satisfaction with MIST (*p* < 0.05). Similarly, six RCTs reported that patients in the MIST group presented significantly less post-surgical pain and lower wound healing scores (*p* < 0.01). *Conclusions*: It was concluded that using MIST resulted in more clinical studies reporting better clinical outcomes. Considering aesthetic appearance, slightly more than half of the clinical trials also showed improved results with MIST. Likewise, regarding postoperative morbidity, 60% of the clinical trials also described better scores with MIST. All of this indicates that MIST is a good alternative for the management of soft tissues.

## 1. Introduction

Dentofacial aesthetics are an aspect of great importance for patients. For this reason, they look for treatment modalities that help to improve their dentofacial aesthetics when the aesthetic proportion of the face and teeth are relevant, not only in dental procedures but also in cosmetic surgery [1,2]. Thus, minimally invasive therapeutic modes have converted the standard of care in numerous medical and dental specialties [3].

The necessity for less invasive and patient-friendly surgical therapies guided the development of the “minimally invasive surgical technique” (MIST) [4]. Consequently, MIST has been incorporated in the diminution of surgical flap extent alongside papilla preservation to increase wound healing and decrease morbidity, and improved results have been observed in clinical studies [5,6]. MIST for excessive gingival display, ridge augmentation, and minimally invasive flapless implant placement have been also reported with predictable outcomes [3,7,8,9]. Similarly, MIST has been used for the treatment of gingival recessions, achieving predictable root coverage [10,11,12].

In this context, MIST has converted common procedures and is recognized by clinicians and patients for its post-treatment benefits and diminution of chair time in comparison to conventional surgeries. Moreover, MIST has been used alone or together with biomaterials/biologicals (graft biomaterials and growth factors) in clinical studies [13,14,15]. Compared to open surgical procedures, MIST involves minor clinical trauma, reduced operating times, prompt post-surgical recovery, fewer post-surgical difficulties, and increased patient comfort [16,17,18].

On the other hand, patient-associated results are one of the main preoccupations in the choice of surgical management of soft tissues. Therefore, approval of patients’ aesthetic needs should be contemplated for successful soft tissue management [19,20]. Thus, the American Academy of Periodontology recommends considering the results reported by the patient to make relevant clinical decisions [21].

Three systematic reviews have evaluated the efficacy of MIST [22,23,24]. Two were oriented to the treatment of intraosseous defects [22,23] and one was directed merely to the management of gingival recessions [24]. Only one of these three reviews evaluated the importance of aesthetic results and postoperative morbidity in the treatment of gingival recessions [24]. Moreover, seeing that MIST has been used in the management of soft tissues and not only in root coverage, it is also relevant to carry out a review in this regard. Considering the importance of the assessment of cosmetic results and postoperative morbidity with MIST, this systematic review aimed to evaluate not only the clinical efficacy of MIST but also to consider its aesthetic results and postoperative morbidity.

## 2. Materials and Methods

This analysis of randomized clinical trials was carried out considering the PRISMA guidelines (Preferred Reporting Items for Systematic Reviews and Meta-analyses) [25]. Moreover, this protocol was registered in PROSPERO (International Prospective Register of Systematic Reviews-receipt 423277). Several databases, including SCOPUS, PubMed/MEDLINE, SCIELO, LILACS, and ClinicalTrails.gov, as well as the gray literature were used in the review structure. Up to and including January 2023, searches were conducted using keywords and MeSH phrases to find publications in all languages, incorporating the following terminologies: minimally invasive surgery, minimally invasive surgical procedures, minimal access surgical procedures, periodontal plastic surgery, perio-esthetics, dental aesthetic, flapless, aesthetics, surgical procedures, plastic, microscopy, and randomized clinical trials issued in all languages. The next exploration procedure was utilized to search the databases using Boolean operators (AND, OR): “minimally invasive surgical procedures” OR “minimally invasive surgery” OR “periodontal plastic surgery” OR “dental aesthetic” AND “flapless” AND “perio-esthetics” OR “minimally invasive surgery.”

The selection criteria of the assessed studies are presented in Table 1.

### 2.1. Questions

As described below, three PICO questions were addressed in this systematic review:

In patients requiring soft tissue management, does a MIST protocol improve clinical outcomes when compared to the same surgical approach but performed under a non-MIST protocol?

In patients requiring soft tissue management, does a MIST protocol present better aesthetic results when compared to the same surgical approach but performed under a non-MIST protocol?

In patients requiring soft tissue management, does a MIST protocol present better postoperative morbidity results when compared to the same surgical approach but performed under a non-MIST protocol?
P = patients requiring soft tissue management;I = MIST protocol;C = non-MIST protocol;O = clinical outcomes, aesthetic results, postoperative morbidity.

### 2.2. Review Process

To determine possible eligibility, two researchers looked over the titles and abstracts and chose randomized clinical studies. If there was a difference of opinion among the authors, randomized clinical trial eligibility was decided by consensus. The statistical test Kappa was employed to evaluate the importance of observer agreement (>90).

### 2.3. Data Collection

The most pertinent information from the chosen randomized clinical trials was included in a table. Each researcher carried out this procedure on their own. The data were then contrasted. Authors’ names, country, date of publication, age of participants, intervention procedure and control group, number of participants, diagnosis, design of the study, relevant methodological aspects, aesthetic perception of patients, postoperative morbidity, study limitations, and length of the follow-up period were all included in the recorded data.

### 2.4. Risk of Bias

Two authors (CMA and AMVB) evaluated the quality and risk of bias of the included clinical trials using a scale for randomized clinical trials [26]. This scale assessed the inclusion of randomization and double blinding and that these were appropriate, as well as the description of the withdrawals.

## 3. Results

The online search turned up 361 studies. Following an examination of the titles and abstracts, 10 duplicate papers and 322 studies were eliminated due to their lack of relevance. Eighteen more studies were eliminated after reading the complete text because they failed to meet several requirements for selection. Ultimately, this systematic review comprised 11 randomized clinical trials [3,7,10,19,27,28,29,30,31,32,33] (Figure 1).

Table 2 presents the diagnosis, design of the study with the intervention carried out, relevant methodological aspects, clinical results of the evaluated study, aesthetic results of the intervention, postoperative morbidity, study limitations, and follow-up time of each clinical trial.

These randomized clinical trials were published between 2005 and 2022. The experiments evaluated 273 patients with a minimum sample of 7 patients [19] and a maximum sample of 40 [7,27], followed by a period of between 4 months [7] and 2 years [28,32]. These trials assessed MIST for papilla preservation [27,28], excessive gingival display [3], minimally invasive flapless implant placement [7], and treatment of gingival recessions [10,19,29,30,31,32,33]. Most of the clinical trials had a parallel [7,27,29,31] or split-mouth design [3,10,19,32,33]. Moreover, three studies used biomaterials [27,28,30].

Herein, the studies used different clinical assessment methods, but most considered traditional periodontal parameters such as plaque indices, bleeding on probing, probing depth, and clinical attachment level. The clinical trials also used questionnaires and analogous visual scales to assess aesthetic results and postoperative morbidity (Table 2).

Table 2 depicts that the two studies that explored MIST for papilla preservation presented greater efficacy in increasing papillary height (*p* < 0.05) [27,28]. However, there were no differences in terms of periodontal parameters [27]. The open flap and minimally invasive flapless techniques showed stable and comparable clinical outcomes for up to 12 months for the management of excessive gingival display [3]. Similarly, one randomized clinical trial described equivalent clinical results using the minimally invasive flapless technique for single implant placement in comparison to flapped implant surgery [7]. Considering the treatment of gingival recessions, contradictory results were observed. While four randomized clinical trials presented greater root coverage with MIST (98–99%; *p* < 0.05) [10,26,27,33], three randomized clinical trials did not show differences between groups [19,29,31]. Furthermore, one experiment described that MIST presented a significant difference in the ultrasonographic thickness of gingiva (*p* < 0.003) [30].

Considering the aesthetic perception of patients, the findings were also divided (Table 2). Four randomized clinical trials indicated high patient satisfaction with MIST for root coverage and papillary augmentation (*p* < 0.05) [10,19,28,30], and one trial described an ideal result from an aesthetic point of view using MIST [33]. Instead, three randomized clinical trials reported great and similar satisfaction for root coverage in the patients of both groups [29,31,32]. Similarly, one randomized clinical trial reported equal manifestations of satisfaction in the test and control groups during the treatment of excessive gingival display (*p* > 0.05). However, in this clinical trial, the researchers perceived that MIST allowed more rapid healing and inferior tissue swelling after one week, in comparison with the traditional method, specifically in the papilla regions where residual scars were observed in the non-MIST patients [3]. On the other hand, two studies did not present results related to aesthetic perception [7,27].

Responding to the third question of this systematic review (Table 2), four randomized clinical trials described that the patients reported low morbidity for MIST and the control treatments [3,10,28,31]. However, five other clinical trials reported that patients in the MIST group presented significantly less post-surgical pain and lower wound healing scores (*p*< 0.01) [7,19,29,30,32]. In addition, one trial indicated that a minimally invasive method might result in less tissue trauma [33]. On the other hand, one randomized clinical trial reported no residual or new sensitivity in the MIST group after 6 and 12 months. Meanwhile, some patients in the control group continued to complain about hypersensitivity at these assessment times [10]. Inversely, another clinical trial indicated that both groups presented a considerable reduction in dentinal hypersensitivity from baseline up to 6 months, but with no significant disparity among groups [29]. One clinical trial did not present results related to postoperative morbidity [27].

Finally, one randomized clinical described that the complete operative time was comparatively inferior (*p* = 0.007) with MIST over non-MIST therapy [19]. Likewise, another clinical trial showed that minimally invasive surgery resulted in a 25% diminution in clinical time, in comparison to conventional surgery [3]. Nevertheless, three randomized clinical trials reported that the mean time consumed in procedures with MIST was longer [10,32,33].

Two randomized clinical trials were of low quality, while the remaining nine were of good quality (Table 3).

However, it is crucial to point out that the trials covered in this review displayed significant heterogeneity in their designs, as evidenced by the use of various therapies, wide variation in patient study characteristics, and variability in the results, among other aspects. Due to these factors, conducting a quantitative analysis is challenging.

## 4. Discussion

To lessen invasiveness, medical and dental surgical procedures have undergone significant changes [8,9]. At the same time, new equipment and materials have been created in preparation for the inevitable advancement of the surgical arsenal. This creative strategy has fortunately been added to the field of periodontal surgery [10,11]. Minimally invasive surgery aims to handle the soft and hard tissues gently and leave behind few incisions and flap reflections [9,11]. Aspects of the wound, blood clot stability, and primary wound closure for blood clot protection were stressed by Cortellini and Tonetti [13] with the minimally invasive surgical technique. The term “minimally invasive surgery” refers to the use of surgical operations that are smaller, more precise, and are made feasible using magnifying tools such as operating microscopes and microsurgical tools and materials [9,10,11,13]. The use of operating microscopes or magnifying lenses, as well as microsurgical tools and materials, has tremendously aided the development and improvement of MIST methods [4,12,13]. Cortellini and Tonetti [13] suggested using an operating microscope in periodontal regenerative surgery, reporting an improved ability to manipulate the soft tissues that resulted in an improved potential for primary wound closure from an average of 70% obtained with regular surgery to an excellent 92% obtained with microsurgery. Some authors have observed better results when employing operating microscopes in various periodontal surgical procedures, from flap surgery to mucogingival surgery [9,10,11].

The efficacy of MIST for the management of soft tissues was assessed in this systematic review of randomized clinical trials for the first time, considering aesthetic results, postoperative morbidity, and clinical outcomes. The three issues asked in this systematic review were related to these subjects.

Previously, one review mainly evaluated the clinical efficacy of MIST for the treatment of gingival recessions [24]. Some limitations of that review are noteworthy. All types of studies, including case series, were considered. Furthermore, smokers were also involved. Only two of the included studies reported cosmetic outcomes and postoperative morbidity [32,34]. Some indices of heterogeneity were high to perform meta-analyses with reliable results. As the authors themselves recognized, all of this reduced the level of evidence [24].

Herein, the randomized clinical trials that explored MIST for papilla preservation presented greater efficacy in increasing papillary height (*p* < 0.05) [27,28]. Very few publications with pre- and post-therapy evaluations of surgical papilla augmentation were found [35]. In contrast to the papillary height gained in the randomized clinical trials reviewed here, a clinical study that used conventional techniques found that the diminution achieved between the contact point and the gingival margin was not statistically significant [35]. The authors of one of the experiments evaluated in this review indicated that the success obtained in the regeneration of the papilla was due to the use of MIST [27]. Utilizing MIST, it was observed that when the concentrated growth factor was positioned subjacent to the flap, it was feasible to recreate a renewed papilla that provided appropriate support around the soft tissues [27].

On the other hand, the open flap and minimally invasive flapless techniques showed stable and comparable clinical outcomes for up to 12 months for the management of excessive gingival display [3]. Moreover, at 12 months, the gingival margin reduction reached 1.0 mm for both techniques. These clinical outcomes are comparable with those of research establishing that the variations in the gingival margin from those demarcated after a standard open-flap crown lengthening were minimal at 6 months [36].

Herein, one randomized clinical trial described similar clinical results using the minimally invasive flapless technique for single implant placement in comparison to flapped implant surgery. Less soft tissue swelling did not cause a positive effect on the preservation of marginal bone around the implant with MIST in the primary healing stage, in comparison to the traditional open flap procedure, [7].

Considering the treatment of gingival recessions, contradictory results were observed. While four randomized clinical trials presented greater root coverage with MIST [10,26,27,33] and one described greater gingival thickness with this procedure [30], three trials did not show differences between groups [19,29,31]. The successful outcomes of the MIST arm, in the experiments evaluated here [10,32,33], were comparable to those described previously in clinical studies that implemented operative microscopes to accomplish gingival recession therapy [33,37]. A feasible justification for the greater average root coverage and occurrence of complete root coverage when MIST is performed is the improved visual perception after the amplification and enhanced illumination of the operative field [24,38]. Moreover, these benefits are related to the utilization of specially designed microsurgical implements and permit a more precise, atraumatic management of the soft tissue, resulting in better healing [39]. On the other hand, the results reported by the other randomized clinical trials reviewed here that did not show differences between groups [19,29,31] agreed with the preceding conclusions describing the utilization of coronally advanced flap for single and multiple gingival recessions [40,41]. It is relevant to note that differences in the methodologies used to measure the clinical parameters make it difficult to compare these investigations.

Regarding the aesthetic perception, five randomized clinical trials revised here indicated better results with the use of MIST. Patients’ expectations concerning root coverage treatments in the aesthetic zone have been growing recently [10,19,28,30]. Thus, it was described that patient satisfaction with the aesthetic effect was 100% in the MIST group and 79.1% in the comparison group [10]. As has been reported, the incidence of total root coverage can elucidate the disparity in patient satisfaction among the groups [41]. On the other hand, the remodeling of a gummy smile is a relevant factor not only in the aesthetics of the smile but also in self-esteem [3]. The randomized clinical trial that treated patients with excessive gingival display in this review described similar aesthetic results for the two intervention groups; nevertheless, some scars were perceived in the papilla of patients in the control group [3]. Open interproximal spaces may also produce aesthetic concerns. One randomized clinical trial explored here showed that the visual scale score for papilla augmentation was superior for the test sites over the placebo [28]. The outcomes described here confirmed that patient approval was more substantial when the procedure is minimally invasive [19,28,30]. The good aesthetic results caused by MIST have also been described in recent clinical studies [42,43]. Some studies reviewed here unfortunately did not report information related to aesthetic results [7,27]. Therefore, the information provided by the remaining nine studies was considered. This indicates the need to promote protocols for studies to consider this very important aspect.

Responding to the third question of this systematic review, four randomized clinical trials described that the patients reported low morbidity for MIST and the control treatments [3,10,28,31]. However, six other randomized trials reported that patients in the MIST group presented significantly less post-surgical pain and lower wound healing scores (*p* < 0.01) [7,19,29,30,32,33]. Dissimilarities in postoperative guidelines, type of analgesic prescription, procedures utilized to remove the graft from the palate, and clinician practice can clarify the disparities between the reports [44]. Furthermore, the inferior postoperative discomfort reported for some trials might be ascribable to the implementation of MIST. Regarding hypersensitivity, it has been described that the root zone close to the cementoenamel is the most vulnerable [45]. Consequently, the disparity between treatments concerning the occurrence of complete root coverage can elucidate the residual hypersensitivity reported in some cases [10]. Only one trial did not describe information related to postoperative morbidity [27]. Thus, the information presented by the remaining 10 clinical trials was pondered. Again, this highlights the need to include postoperative morbidity in recommended guidelines for describing clinical trials.

On the other hand, the frequency and harshness of difficulties and pain after surgery correlate adequately with the extent of the procedure [46]. Although the results related to surgical times were variable in this review, it has been described that long surgical times can compensate for the favorable therapy results of MIST [23,33].

Eleven randomized clinical trials were included in this systematic review due to the tight screening criteria. However, some study limitations were present in this review. Two randomized clinical trials that were considered for this systematic review had a significant risk of bias (most bias was due to double blinding) [30,32]; nevertheless, the other randomized clinical trials that were selected were of high quality [3,7,10,19,27,28,29,31,33]. In any event, to confirm the findings presented here, higher quality clinical trials are needed. Unfortunately, two studies did not report information related to aesthetic aspects [7,27] and one study did not report postoperative morbidity [27], which prevented more robust analyses and hence we relied on the remaining clinical trials. In addition, some clinical trials had small sample sizes [19,29,33] and short follow-up periods [28]. It is important to note that the heterogeneity of the randomized clinical trials examined in this systematic review precluded a more thorough study. Some recent systematic reviews [47,48] have revealed the exact same issues. The diverse behavior of systematic reviews, in terms of the clinical trials included, justifies the standardization of clinical techniques so that comparisons between clinical trials may be made without prejudice. On the other hand, the fact that the included randomized clinical trials were longitudinal in nature should be emphasized as a strength of this systematic review. However, further clinical studies are needed to assess the efficacy of MIST in treating soft tissue, particularly when postoperative morbidity and aesthetics are considered.

## 5. Conclusions

Within the limitations of this current systematic review, it was concluded that more clinical trials report better clinical outcomes when MIST is used. Considering the aesthetic appearance, slightly more than half of the clinical trials showed improved results with MIST. Similarly, regarding postoperative morbidity, 60% of the clinical trials described better scores with MIST. All of this indicates that MIST is a good alternative for the management of soft tissues.

## Figures and Tables

**Figure 1 medicina-59-00924-f001:**
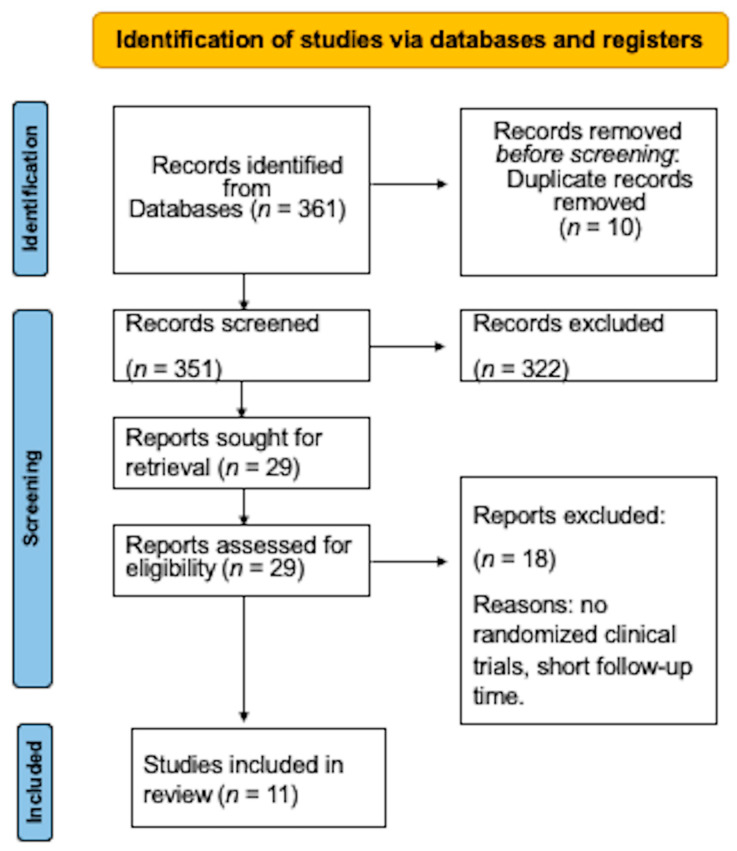
Flowchart of the randomized clinical trial selection method.

**Table 1 medicina-59-00924-t001:** Selection criteria of the assessed studies.

Inclusion Criteria	Exclusion Criteria
Randomized clinical trials (containing experimental group versus control group). Clinical trials with at least 3 months of follow-up. Persons without systemic diseases. MIST to treat different soft tissue conditions.	Patients with any disorder that can disturb healing. Patients who underwent periodontal surgical therapy in the preceding 12 months. Smokers. Continued traumatic tooth brushing. Deficient plaque control procedures. Bruxism and/or parafunctional habits. Patients with orthodontic appliances. Duplicate publications. In vitro experimentations and investigations implemented on animals.

**Table 2 medicina-59-00924-t002:** Features of the randomized clinical trials evaluated.

Authors, Place, Publication Date	Patients /Mean Age	Diagnosis/ Condition	Trial Design/ Experimental versus Control Group	Methods of Clinical Evaluation	Clinical Results	Methods of Aesthetic Evaluation and Postoperative Morbidity	Aesthetic Outcomes	Postoperative Morbidity	Limitations Presented by the Authors	Follow-Up Period
Çankaya et al., Turkey 2020 [27]	40 32 years	Papillary losses in maxillary anterior teeth	Parallel experimental group: 20 patients with 60 multiple adjacent papillary losses receiving minimally invasive surgery with concentrated growth factor. Control group: 20 patients with 60 multiple adjacent papillary losses without surgery.	Plaque index, probing depth, bleeding on probing, measurement of the width of the keratinized gingiva, and papillary thickness were evaluated before surgery and at 3, 6, and 12 months after surgery. Changes in the interproximal papilla and papilla filling were evaluated at 3, 6, and 12 months after surgery. Digital impression images were taken and transferred to software. Three separate images were taken at each session at baseline and 3, 6, and 12 months.	The papillary area at 3, 6, and 12 months showed statistically significant differences from baseline values in the test group (*p* < 0.001) but not in the control group.	NR	NR	NR	MIST is not recommended in cases of insufficient keratinized gingival width, shallow vestibular depth, presence of a high frenulum.	12 months
McGuirre & Scheyer, USA 2007 [28]	20 Age NR	Interdental papillary recession defects	Crossover experimental and control groups: Each subject (20 patients) served as his/her own control by receiving test and placebo therapy (cell culture media). Two primary sites were designated and randomized to receive autologous fibroblast or placebo injections beginning 1 week following the papilla priming method; two additional injections were completed 7 to 14 days following the initial injections.	Percentage changes in papillary height of the primary treatment areas from baseline to the 4-month visit. Changes in the following parameters from baseline to the 4-month visit: distance from the tip of the papilla to the alveolar crest and from the base of the contact area to the alveolar crest, probing depth, interproximal width of papilla, and plaque index. Photographs were taken at a standard magnification. Radiographs and study impressions of sites were taken at baseline and 4 months.	The management zones presented a statistically significant mean percentage increase from baseline in papillary height (*p* = 0.0067) at 2 months. The difference between test and placebo sites in papillary height at 2 months approached statistical significance (*p* = 0.006), recommending that the test therapy was greater than placebo management.	Inflammation score, tissue texture and color, and patient and clinician perception of change in the Nordland Class Score. A visual analog scale was used by the examiner and subject to assess the defect change from baseline to 2, 3, and 4 months. Assessment of subject safety included an analysis of the incidence of adverse events.	The examiner and subject’s visual scale scores were statistically significantly different from baseline for both treatment groups and superior for the test sites over the placebo (*p* = 0.01).	The fact that there were no significant changes in inflammation nor tissue texture, and color following treatment indicated that the therapy was well tolerated and yielded no adverse effects. The treatment was pain-free for both groups.	Small sample size. Angulation for digital photographs was not standardized. Volumetric increase in papilla following treatment was not measured.	4 months
Ribeiro et al., Brazil 2014 [3]	28 28 years	Altered passive eruption	Split-mouth experimental and control groups: Contralateral quadrants received aesthetic crown lengthening using open-flap (28 sides, 105 teeth) or MIST techniques (28 sides, 105 teeth) for the treatment of excessive gingival display.	Plaque index, probing depth, bleeding on probing, gingival margin, clinical attachment level, and keratinized gingiva height were evaluated at baseline and 3, 6, and 12 months post-surgery. Bone level was noted before and after the surgical techniques. The gingival crevicular fluid levels of receptor activator of nuclear factor-kB ligand and osteoprotegerin were assessed at baseline and 3 months post-surgery. To assess the hard and soft tissues, soft-tissue cone beam computed tomography was performed at baseline.	Probing depths were reduced for both groups at all time points, compared with baseline (*p* < 0.05). There were no differences between groups for gingival margin reduction at any time point.	Patient perceptions regarding morbidity and aesthetic satisfaction were evaluated with a questionnaire and the responses were quantified with a visual analog scale. The questionnaire was obtained upon completion of the procedure (pain), at 7 days post-surgery (pain/discomfort, swelling, hematoma, aesthetic appearance), and at 6 months post-surgery (esthetic appearance).	Patients reported high satisfaction with the aesthetic appearance of both procedures.	Patients reported low morbidity for both procedures.	MIST is not recommended in cases of insufficient keratinized gingival width. The reduction of bone in the buccal-palatal direction in cases of thick bone is not possible using MIST.	12 months
Wang et al., China 2017 [7]	40 39 years	Lost mandibular first molar at least 3 months of post-extraction healing.	Parallel experimental group: 20 patients intervened with minimally invasive flapless method for single implant placement. Control group: 20 patients intervened with flapped implant surgery.	Cone beam computerized tomography was taken at the day of implant installation. Modified sulcus bleeding index and plaque index were evaluated at 1, 2, and 4 weeks post-surgery and 3, 6, 12, and 24 months post-crown delivery. Probing depth was assessed at 4 weeks post-implant insertion surgery, on the day of crown delivery and at 3, 6, 12, and 24 months following intervention. The width of keratinized mucosa was measured between the soft tissue margin and the mucogingival junction at the facial aspects of the abutment on the day of crown delivery and at 12- and 24-month follow-up. Periapical radiographs were completed on the day after implant insertion, crown delivery and at 3-, 12-, and 24-month recall. All images were scanned and transferred to a computer with an image analysis package.	At each appointment, no changes in probing depth and marginal bone loss were observed between groups (*p* < 0.05).	Wound healing index was evaluated at 1, 2, and 4 weeks post-surgery. At 2 weeks post-surgery, post-surgical pain was measured on a visual analog scale by questioning the patient to evaluate their pain after surgery.	NR	Patients in the MIST group described significantly less post-surgical pain (*p*< 0.01) and significantly lower wound healing index scores at 1-week follow-up.	Computer-guided template was not performed for patients. Large sample size and histological analysis are required to confirm the findings.	24 months
Bittecourt et al., Brazil 2012 [10]	24 34 years	Gingival recessions	Split-mouth experimental and control groups: 24 patients in which subepithelial connective tissue graft was performed with a microscope or without a microscope or any type of magnification in the treatment of gingival recessions.	Recession height, weight of keratinized tissue, recession width, probing depth, clinical attachment level, and thickness of the keratinized tissue were registered at baseline and 6 and 12 months post-surgery.	The mean proportions of root coverage for test and control groups after 12 months were 98.0% and 88.3%, correspondingly (*p* < 0.05).	Overall postoperative pain was also assessed using a horizontal visual analog scale. At 6 months, a questionnaire was given to each patient. The questionnaire recorded the results of the procedures relative to aesthetics, root sensitivity (before and after surgery), and the postoperative period.	In the test group, all patients were pleased with the aesthetics achieved, and 19 patients (79.1%) were satisfied in the control group.	For postoperative morbidity, 14 patients in each of the two therapy groups did not take analgesics for pain control.	A longer follow-up period is necessary to verify the stability of MIST. The randomization method and impossibility of masking patients to the use of the microscope.	12 months
Rajendran et al., India 2018 [19]	7 30–48 years	Gingival recessions	Split-mouth experimental and control groups: 7 patients were treated with a minimally invasive coronally advanced flap or with modified coronally advanced flap for the treatment of multiple adjacent gingival recessions.	Recession heigh, recession width, probing depth, clinical attachment level, width of keratinized tissue, and gingival tissue thickness were recorded at baseline and 6 months post-operation.	No disparities were presented among minimally invasive coronally advanced and modified coronally advanced flap places, in the change in gingival recession depth, gingival recession width, clinical attachment level, width of the keratinized tissue, mean, and complete root coverage after 6 months (*p* > 0.05).	For patient-reported outcomes, a questionnaire and visual analogue scale were used. The questionnaire consisted of 2 parts: the first part was regarding aesthetic concerns about the recession, and the second part was regarding the preferred method of treatment among the 2 techniques used in the study. Patient satisfaction with aesthetics was evaluated at 3- and 6-month follow-up visits. The postoperative morbidity was evaluated 1 week after surgery.	The patient-reported aesthetic result was statistically significant (*p* < 0.001) between the minimally invasive coronally advanced flap and the modified coronally advanced flap arms, with better results for the minimally invasive coronally advanced flap arm.	The patient-reported outcome of postoperative morbidity was statistically significant (*p* < 0.001) between the minimally invasive coronally advanced flap and the modified coronally advanced flap sides, with better results on the minimally invasive coronally advanced flap side.	Small sample size. Studying various classes of Miller recession defects was recommended.	6 months
Karmakar et al., India 2022 [29]	10 44 years	Gingival recessions	Parallel experimental group: 5 patients treated with modified microsurgical tunnel technique utilizing connective tissue graft in the coverage of multiple adjacent recessions. Control group: 5 patients treated with modified coronally advanced flap utilizing connective tissue graft in the coverage of multiple adjacent recessions.	Sulcular bleeding index, recession depth, probing depth, clinical attachment level, keratinized tissue width, and gingival biotype were recorded at baseline and 1, 3, and 6 months post-surgery.	Mean root coverage and complete root coverage for the experimental group were 92.01% and 80% (*p* = 0.703) and for the control group were 87.39% and 60% (*p* = 0.545).	At baseline and 1, 3, and 6 months post-surgery, patients were provided with a questionnaire to subjectively evaluate their dentinal hypersensitivity. Quantitative evaluation was performed using a visual analog scale. Patient morbidity was assessed by subjective evaluation from the patient regarding pain, bleeding, and swelling 7 days after the surgery. The aesthetic evaluation was performed using the Root Coverage Aesthetic Score by comparing the digital images taken at baseline and 6 months by the operator.	Both therapies described high aesthetic outcomes.	Patients in the control group presented more morbidity (*p* < 0.05).	A longer follow-up period is necessary.	6 months
Srivastava et al., India 2021 [30]	30 Age NR	Gingival recessions	Parallel experimental group: 15 patients with recession defects were managed with a coronally positioned flap and acellular dermal matrix utilizing microsurgery. Control group: 15 patients with recessions were managed with a coronally positioned flap and acellular dermal matrix applying a conventional method.	Height of gingival recession, probing depth, clinical attachment level, gingival thickness, and width of the attached gingiva were documented at baseline and 3 and 6 months.	The MIST technique exhibited a significant change in the ultrasonographic thickness of gingiva (*p* < 0.003).	At 10 days and 1 and 6 months post-operation, patient satisfaction was recorded on a scale of 1–10. The satisfaction criteria included intra-operative experience at 10th day (pain during surgery and discomfort experience related to the duration of procedure and handling by the operator), postoperative experience at 1 month (for pain, swelling and postoperative complications), hypersensitivity at 6 months, recession coverage at 6 months, and appearance (color and form) at 6 months.	The MIST technique confirmed an improved patient satisfaction result (*p* < 0.005).	Postoperative morbidity was better in the experimental group (*p* < 0.005).	NR	6 months
Azaripour et al., Germany 2016 [31]	40 39 years	Gingival recessions	Parallel experimental group: 20 patients were treated with the coronally advanced flap and the modified microsurgical tunnel technique for the management of recessions. Control group: 20 patients were treated with the coronally advanced flap for the management of recessions.	Clinical measurements and volumetric evaluation of the soft tissue and digital photographs were performed at baseline and 1, 3, 6, and 12 months after surgery. The following clinical measurements were performed: probing depth, recession of the gingival margin, and width of keratinized tissue. The evaluation based on the Root Coverage Aesthetic Score was performed by comparing the corresponding images that were taken at baseline and 6- and 12-month reevaluation appointments.	Root coverage was 98.3% for the coronally advanced flap and 97.2% for the modified microsurgical tunnel technique (*p* > 0.05)	The aesthetic outcomes were evaluated using the Root Coverage Aesthetic Score. The evaluation compared 3 corresponding images of each experimental unit taken at baseline and 6 and 12 months. Immediately after surgery and again after 2 weeks, a questionnaire was given to the patients for subjective evaluation of the treatment in terms of pain, fear, morbidity, overall satisfaction, and root sensitivity. The parameters were evaluated quantitatively using a visual analogue scale. At the 1-year evaluation, patient satisfaction and their willingness to undergo further periodontal surgery were explored.	Both treatments described high aesthetic results (9.2 ± 1.3 for the coronally advanced flap and 9.2 ± 1.1 for the modified microsurgical tunnel technique; *p* > 0.05).	Both therapies showed certain post-surgical pain. On a measure from 0 to 10, the noticed pain was 2.2 ± 2.9 for the the coronally advanced flap test group and 2.8 ±2.9 for the modified microsurgical tunnel technique group (*p* > 0.05).	NR	12 months
Nizam et al., Turkey 2015 [32]	24 27 years	Gingival recessions	Split-mouth experimental and control groups: 21 teeth in microsurgical technique (experimental group) and 21 teeth in macrosurgical treatment (control group) were managed by implementing coronally positioned flap and subgingival connective tissue graft in the coverage of gingival recessions	Silicone impressions were made, and stone models of each defect were obtained. Photographs were also taken (preoperative procedure). Plaque index, gingival index, and probing depths were recorded. Recession depth, recession width, keratinized tissue width, and root surface area of the recessions were calculated. All clinical measurements except probing depth were made on standardized photographs. Clinical attachment level was also measured. These parameters were evaluated at baseline, 1, 3, 6, and 24th months.	Recession depth and recession surface area at 24 months were significantly lower in the microsurgical group (*p* < 0.05).	Postoperative pain of the intervention was evaluated using a visual analog scale during the first week. The patients also assessed the aesthetic result at baseline and 3, 6, and 24 months using a visual analog scale.	Aesthetic results improved likewise in both treatments.	The pain results in the donor and recipient zone diminished earlier in the microsurgical group (*p* < 0.05).	The calculation of root coverage percentage based on recession depth may result in overestimation of root coverage percentage and therefore could also be validated using root surface area values.	24 months
Burkhardt & Lang, Switzerland 2005 [33]	10 32–44 years	Gingival recessions	Split-mouth experimental and control groups: 10 patients with bilateral Class I and II recessions at maxillary canines were randomly selected for recession coverage either by a microsurgical (experimental group) or macrosurgical (control group) technique.	Clinical examinations at the recession sites were carried out before the surgical procedures and then after 1, 3, 6, and 12 months post-operation. The following parameters were assessed: gingival and plaque index, probing depth, clinical attachment level, and gingival recession. The percentage of vascularization was analyzed on the standardized angiographic images in defined areas of the gingival surfaces (obtained immediately after surgical intervention and 3 and 7 days post-operation).	The clinical lengths showed a mean recession coverage of 99.4% for the experimental and 90.8% for the control sites after the first month of healing (*p* < 0.05). The proportion of root coverage in both test and control sites persisted unchanging through the first year at 98% and 90%, correspondingly. The vascularization of the grafts was significantly improved by the microsurgical method (*p* < 0.05).	Subjective point of view	Complete root coverage was the ideal treatment outcome from an aesthetic and subjective standpoint. After one month, 90% of the test sites and 40% of the control sites had total root coverage, respectively (*p* < 0.05).	The microsurgically operated sites had greater vascularization than the macrosurgically treated sites, based on the angiographic study carried out immediately after the surgical procedure (*p* = 0.02). The distinction revealed proof that a minimally invasive method might result in less tissue trauma.	NR	12 months

NR = Not reported.

**Table 3 medicina-59-00924-t003:** Quality of the selected randomized clinical trials [26].

Randomized Clinical Trial	Randomization	Double Blinding	Withdraw	Proper Randomization	Proper Double Blinding	Score
Çankaya et al., 2020 [27]	1	0	1	1	0	3
McGuirre & Scheyer, 2007 [28]	1	1	1	1	1	5
Ribeiro et al., 2014 [3]	1	0	1	1	0	3
Wang et al., 2017 [7]	1	0	1	1	0	3
Bittencourt et al., 2012 [10]	1	0	1	1	0	3
Rajendran et al., 2018 [19]	1	1	1	1	1	5
Karmakar et al., 2022 [29]	1	1	1	1	0	4
Srivastava et al., 2021 [30]	1	0	1	0	0	2
Azaripour et al., 2016 [31]	1	0	1	1	0	3
Nizam et al., 2015 [32]	1	0	1	0	0	2
Burkhardt & Lang, 2005 [33]	1	0	1	0	0	3

## Data Availability

The data obtained in this review were pooled from the included investigations.

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
