# Peer review of "A Systematic Review of Randomized Clinical Trials Evaluating the Efficacy of Minimally Invasive Surgery for Soft Tissue Management: Aesthetics, Postoperative Morbidity, and Clinical Results"

_medicina, 2023, doi:10.3390/medicina59050924_

Round 1

Reviewer 1 Report

The authors through this systematic review aimed to evaluate the efficacy of MIST for soft tissue management considering aesthetic, postoperative morbidity, and clinical results finding that MIST showed good aesthetic results and less post-28 operative morbidity compared to conventional procedures.

They state that the review was done with "PRISMA in mind". please clarify what "in mind" does it mean?

Figure 1 has low quality, please correct that.

Table 2. The information contained in the table is not critical for the reader. Please consider removing it.

Most paragraphs are written without structure. Please correct the paragraphs adding at the beginning the main idea and then developing the idea along the paragraph.

Authors should improve English since the redaction does not sound technically correct.

Author Response

Responses Reviewer 1

Dear Reviewer,

We are grateful for the constructive comments you provided, which helped us to improve the manuscript significantly.

Our responses to your comments are outlined below and highlighted in yellow (to differentiate them from responses to other reviewers) in the new version.

1. They state that the review was done with "PRISMA in mind". please clarify what "in mind" does it mean?

RESPONSE: The sentence was amended.

2. Figure 1 has low quality, please correct that.

RESPONSE: The figure was improved.

3. Table 2. The information contained in the table is not critical for the reader. Please consider removing it.

RESPONSE: The risks of bias should be presented according to the PRISMA guideline. All the information presented there is important for readers to visualize where the biases of the studies are located. Furthermore, presenting all this information in the main text is complex.

4. Most paragraphs are written without structure. Please correct the paragraphs by adding at the beginning main idea and then developing the idea along the paragraph.

RESPONSE: We try to do our best to follow your recommendations.

Reviewer 2 Report

Dear Author

Please find the comments attached 

Regards

Its Ok

Author Response

Responses Reviewer 2

Dear Reviewer,

We are grateful for the constructive comments you provided, which helped us to improve the manuscript significantly.

Our responses to your comments are outlined below and highlighted in blue (to differentiate them from responses to other reviewers) in the new version.

1. It is mentioned that- morbidity, then, improved periodontal parameters have been observed in descriptive studies.. kindly revise.

RESPONSE: The observation was reviewed and corrected.

2. It is mentioned that- MIST has converted common and is recognized by clinicians and patients due to post-treatment wellness,.. kindly revise

RESPONSE: The observation was reviewed and corrected.

  1. It is mentioned that- Two are oriented to the treatment of intraosseous defects and one is directed only to the management of gingival recessions.. kindly cite the references accordingly

RESPONSE: The observation was reviewed, and the references were cited accordingly.

  1. It is mentioned that- Two of these three reviews superficially assessed such important aspects as aesthetic perception.. kindly revise

RESPONSE: The observation was reviewed and corrected.

  1. Kindly mention the detailed aim of the study at the end of the introduction

RESPONSE: The objectives were presented at the end of the introduction.

  1. It is mentioned that- This analysis of randomized clinical trials was done with the PRISMA in mind.. kindly revise

     RESPONSE: The observation was reviewed and corrected.

  1. Kindly mention the initial search date considered too in the methodology.

RESPONSE: “Up to and including January 2023” indicates that there is no exact initial search limit. This avoids selection bias.

  1. Kindly address whether PICO was considered.

RESPONSE: Three PICO questions were asked in this review. Given the concerns of the reviewer, the questions were defined in more detail.

  1. As RCT studies were considered- kindly mention if any comparators were considered

RESPONSE: All included clinical trials had an experimental group and a control group.

  1. As all languages were considered, kindly comment whether interpretation was possible by the reviewers.

RESPONSE: All included studies were published in English; therefore, no interpretation was required. In any case, resources were considered in the case of including studies in languages other than English and Spanish.

  1. It is mentioned that- quality and risk of bias of the included clinical trials using a scale for randomized clinical trials.. kindly enumerate.

RESPONSE: The scale had already been referenced with the number 26. The elements that compose it were detailed.

  1. It is mentioned that- aesthetic perception of patients, postoperative morbidity, as well as the length of the follow-up period were all included in the recorded data… kindly comment on the clinical results too as stated in the title

RESPONSE: Suggested comment was included

  1. Dear authors, kindly comment as MIST can be considered for treating different conditions and in procedures, kindly comment whether an array of procedures was considered or was a particular interest considered.

RESPONSE: This observation was included in the selection criteria.

  1. Flowchart Comments.

RESPONSE: We use the flow chart recommended by the PRISMA guide as presented on the website. The PRISMA guide does not indicate the need to mention the arrows between the boxes since each box clearly indicates its content. The rest of the recommendations were followed.   

  1. Characteristics table can further be structured. Inclusion of columns can be considered.
  2. Kindly mention the diagnosis/ condition/ reason for the procedure.

RESPONSE: This observation was included in the table.

  1. Kindly mention the sample distribution as even controls may be considered – hence for test and control- method/ procedure considered for control too.

RESPONSE: This observation was included in the table.

Kindly enumerate the study design if split mouth design were considered.

RESPONSE: This observation was included in the table.

Kindly mention a separate column if biomaterials were considered or were not considered.

RESPONSE: This observation was included in the text.

Kindly enumerate the evaluation factors in general considered including aesthetic, morbidity prior to the brief procedure.

RESPONSE: This observation was included in the table.

Kindly mention the method of evaluation for the factors analyzed – the scales/ questionaries, which ever applicable prior to the results

RESPONSE: This observation was included in the table.

Kindly mention the follow up period prior to the results enumeration.

     RESPONSE: This observation was included in the table.

Kindly mention a separate column at the end mentioning the inference and limitations from authors point of view.

RESPONSE: This observation was included in the table.

Kindly mention the place too in the authors column

RESPONSE: This observation was included in the table.

  1. Kindly structure and provide a detailed characteristics table.

 RESPONSE: This observation was included.

  1. Kindly comment on the characteristics table- as few studies are stated to have NR the factors for analysis which forms the objective of the study- especially- Burkhardt & Lang. kindly comment how were they considered for analysis for the present SR as the objectives were delineated.

RESPONSE: These considerations were presented in the results and considered in the discussion.

  1. It is mentioned that- MIST for papilla preservation [27,28], excessive gingival display [3], minimally invasive flapless implant placement [7], and treatment of gingival recessions….kindly note that the table has to represent the former effectively.

RESPONSE: Initially, the table was organized considering the year of publication of the studies. With the observations indicated, the table was organized according to how the results were reported.

  1. It is mentioned that- MIST for papilla preservation [27,28], excessive gingival display [3], minimally invasive flapless implant placement [7], and treatment of gingival recessions…kindly revise. kindly comment as few articles considered do not address the same as presented in characteristics table. Kindly verify.

RESPONSE: With the observations indicated, the table was organized.

  1. Kindly cite the authors in the discussion wherever applicable.

RESPONSE: The recommendation was followed.

  1. Kindly mention the limitations of the present study at the end of the discussion.

RESPONSE: The limitations are presented at the end of the manuscript.

  1. It is mentioned that- efficacy like conventional procedures; however, good aesthetic results, and less postoperative morbidity were observed whit MIST… kindly revise.

RESPONSE: Considering your relevant comments, we review again and in detail all the RCTs and observed that more studies report better clinical results (77% versus 33%). Similarly, the majority reported better esthetic results (56%) and postoperative morbidity results (60%) while some clinical trials reported similar esthetic (44%) and postoperative morbidity (40%) results. Therefore, we decided to present the conclusions in that sense.

Reviewer 3 Report

This is well written paper. 

1. Line 26- Please rephrase as its not clear 

2. Line 59 is not clear- please rephrase 

3. Line 61 to 65 need to be rephrased to clearly understand what prior reviews addressed and what was the gap. 

4. Add a table of exclsuion and inclusion criteria 

5. Add to the table, how did the studies define success 

Overall, paper flows well. The language can be simplied with greater explanantion for the pointe being made. 

Author Response

Responses Reviewer 3

Dear Reviewer,

We are grateful for the constructive comments you provided, which helped us to improve the manuscript significantly.

Our responses to your comments are outlined below and highlighted in green (to differentiate them from responses to other reviewers) in the new version.

1. Line 26- Please rephrase as it’s not clear

RESPONSE: The sentence was rephrased.

2. Line 59 is not clear- please rephrase 

RESPONSE: The sentence was rephrased.

3. Lines 61 to 65 need to be rephrased to clearly understand what prior reviews addressed and what was the gap. 

RESPONSE: The sentence was rephrased.

4. Add a table of exclusion and inclusion criteria

RESPONSE: Everything described in the text, considering the selection criteria, was reconstructed in a table.

5. Add to the table, how did the studies define success

RESPONSE: The observation was added in a new column.

Round 2

Reviewer 2 Report

The revisions are satisfactory 

The revisions are satisfactory 

Author Response

Thanks